# Dispersion-engineered metasurfaces reaching broadband 90% relative diffraction efficiency

Wei Ting Chen[1,4], Joon-Suh Park[1,4], Justin Marchioni[1,2], Sophia Millay[1,3], Kerolos M. A. Yousef[1] & Federico Capasso [1]✉

Dispersion results from the variation of index of refraction as well as electric field confinement in sub-wavelength structures. It usually results in efficiency decrease in metasurface components leading to troublesome scattering into unwanted directions. In this letter, by dispersion engineering, we report a set of eight nanostructures whose dispersion properties are nearly identical to each other while being capable of providing 0 to 2π full-phase coverage. Our nanostructure set enables broadband and polarization-insensitive metasurface components reaching 90% relative diffraction efficiency (normalized to the power of transmitted light) from 450 nm to 700 nm in wavelength. Relative diffraction efficiency is important at a system level – in addition to diffraction efficiency (normalized to the power of incident light) – as it considers only the transmitted optical power that can affect the signal to noise ratio. We first illustrate our design principle by a chromatic dispersion-engineered metasurface grating, then show that other metasurface components such as chromatic metalenses can also be implemented by the same set of nanostructures with significantly improved relative diffraction efficiency.

The demand of compact and high-end devices for sensing, imaging as well as virtual and augmented reality is ever increasing. Photonic metasurfaces, consisting of sub-wavelength spaced nanostructures, are an emerging platform to realize the target performances. Recent advances in high-efficiency metasurfaces have been realized with low-loss dielectrics such as $HfO_2$, $TiO_2$, amorphous silicon and PbTe in the ultraviolet, visible and infrared region[1–5], respectively. Using these optically lossless materials improves efficiency at design wavelengths and has led to various metasurface components such as pulse-shapers[6,7], ultrafast beam deflectors[8–10], metalenses[2,3], depth sensors[11–13], full-Stoke polarization cameras[14,15] and virtual and augmented reality components[16–20]. The manufacturing of these metasurface components, instead of grinding, polishing and molding used in conventional optical components, is based on lithography, the same approach for chip manufacturing. Recent works have demonstrated various metasurface components by industrial deep ultraviolet lithography steppers[21–23] and nano-imprinting for emerging mass-production[24,25]. Such lithography-based manufacturing of optical components is disruptive because a foundry is able to manufacture the whole camera module including the sensor and the optics[26,27]. However, the efficiencies of these metasurface components drop rapidly from typically 80%–90% to ~20% when the incident wavelength shifts from the design wavelength to the edge of a broad bandwidth. The major reason causing the efficiency drop is due to different dispersion properties of the constituent nanostructures[28–30]. Previous efforts aiming to solve this challenge were based on simultaneously using pillar and hole structures[31]; however, this approach results in different pillar heights and hole depths due to the loading effect in dry etching.

Previous dispersion-engineered metasurfaces focused on reducing chromatic focal length shift of metalenses[32–36], correcting

[1]Harvard John A. Paulson School of Engineering and Applied Sciences, Harvard University, Cambridge, MA 02138, USA. [2]University of Waterloo, Waterloo, ON N2L 3G1, Canada. [3]Department of Physics, Williams College, Williamstown, MA 01267, USA. [4]These authors contributed equally: Wei Ting Chen, Joon-Suh Park. ✉e-mail: capasso@seas.harvard.edu

aberrations in hybrid lenses and spectrometers[37,38] and color sorting aiming to replace the Bayer filter[39–41]. Here, we report on how dispersion engineering plays an important role in realizing polarization-insensitive and broadband high relative diffraction efficiency (normalized to the power of transmitted light) metasurface components. In this letter, relative diffraction efficiency is primarily discussed along with diffraction efficiency (normalized to the power of incident light), as the former provides information on how much noise from other diffracted orders will occur at a systems level. We utilize anisotropic nanostructures as a building block, which has more adjustable geometric parameters to fine tune dispersion, leading to a library comprising about 20,000 different nanostructures. The number of nanostructures in our library is significantly larger compared with circular nanopillar libraries used in conventional polarization-insensitive metasurface components. We choose 8 nanostructures out of our anisotropic nanostructure library in such a way that their dispersion is customized to maintain the target phase profile covering nearly the entire visible spectrum. With these selected nanostructures, we first demonstrate a broadband, high relative diffraction efficiency and polarization-insensitive metasurface grating (referred to metagrating hereinafter) with an averaged relative diffraction efficiency up to nearly 90% for wavelengths from $\lambda = 450$ nm to 700 nm. Subsequently, we show that the same set of nanostructures can be used to demonstrate other metasurface components such as chromatic metalenses and beam shapers.

In this letter, relative diffraction efficiency is defined as the power of the +1 grating order divided by the power of transmitted light, while diffraction efficiency is the same but divided by the power of incident light following the terminologies used in ref. 42. For metalenses, relative and absolute focusing efficiencies are defined as the power of focal spot divided by the power of transmitted and incident light, respectively. It is worth mentioning that relative diffraction efficiency multiplied by transmission equals diffraction efficiency (same relation also holds between relative and absolute focusing efficiencies). We emphasize the improvement in relative diffraction and focusing efficiencies so that the 0th order light, ghost image and halos in metasurface components can be reduced. Improvement in transmission will be discussed.

## Results

### Design principle

We start from a metagrating, which is designed to deflect transmitted light to a single +1 order, to illustrate how dispersion plays a dominant role in efficiency over a wide bandwidth. We chose a metagrating as an example since it serves as a building block for many optical devices: metalens, axicons, holograms etc. Therefore, by improving the efficiency of the metagrating, it is possible to improve the efficiency of other metasurface components in general. For comparison, a conventional polarization-insensitive metagrating is shown in Fig. 1a, consisting of circular nanopillars with different diameters. The simulated transmitted wavefronts for three different incident wavelengths (450 nm, 530 nm, and 650 nm) are plotted above the nanopillars, respectively, with their corresponding ideal linear wavefronts plotted in black dotted lines. Each circular pillar can be qualitatively understood as a miniature truncated waveguide, where its effective index $n_{\text{eff}}(\lambda)$ typically increases with diameter and is a function of wavelength $\lambda$. The diameters of these nanopillars were chosen such that, at the design wavelength $\lambda_d$ (here at 530 nm, see the middle wavefront plot in Fig. 1a), they impart a linear phase delay given by:

$$\varphi(x) = \frac{2\pi}{\lambda_d} x * \sin(\alpha) \quad (1)$$

where $x$ is the spatial coordinate at the center of each nanopillar and $\alpha$ is the deflection angle of the 1st order in normal incidence. Their

center-to-center distance is a constant (250 nm) chosen according to our previous publication[43]. Such phase delay leads to high relative diffraction efficiency into the 1st order. However, the linear phase delay starts deviating away from the linear one when incident wavelength increases or decreases from $\lambda_d$ because these nanopillars have different dispersion characteristics due to their effective index dependence on the diameters of the pillars. As shown in the blue-colored plot of Fig. 1a representing the simulated transmitted wavefront at $\lambda = 450$ nm, it is wrinkled and deviates from the ideal linear one (black dotted line) resulting in diffraction to other high orders, lowering the power of 1st order. Moreover, for longer wavelengths such as $\lambda = 650$ nm (the red-colored plot in Fig. 1a), the phase coverage becomes much smaller than $2\pi$, leading to more transmitted power going to the 0th order. Alternatively, one can use nanofins of the same length and width but different rotations to realize a metagrating based on dispersionless Pancharatnam-Berry phase[44–48] to maintain the linear phase delay over a large bandwidth. However, realizing a metagrating in this manner results in polarization-sensitive functionality, i.e., right- and left-handed incident circular polarizations are deflected to a pair of symmetric angles of $\alpha$ and $-\alpha$ respectively. In addition to polarization sensitivity, Pancharatnam-Berry phase metasurfaces still suffer from the decrease of diffraction efficiency over a large bandwidth because the polarization conversion efficiency of a nanofin is a function of wavelength (see Fig. S1 for a comparison against the Pancharatnam-Berry phase metasurfaces of ref. 47).

In sharp contrast, Fig. 1b shows a dispersion-engineered metagrating consisting of anisotropic nanostructures with its simulated wavefronts plotted at blue (460 nm), green (550 nm), and red (650 nm) incident wavelengths, respectively. The wavefronts agree well with the corresponding ideal ones (black dotted lines). Design details and geometric parameters of each nanostructure are given in Figs. S2, S3, respectively. Figure 1c, d are scanning microscope images of a fabricated sample; nanofabrication detail can be found in ref. 47. Although the design uses anisotropic nanostructures, the dispersion-engineered metagrating is polarization insensitive (meaning it diffracts transmitted light to +1 order regardless of incident polarization) because the symmetric axes of its constituent nanostructures are either parallel or perpendicular to each other to cancel the polarization dependent Pancharatnam-Berry phase[49]. For a nanostructure shown in Fig. 1b, its transmitted electric field under circularly polarized incidence $\begin{bmatrix} 1 & \pm i \end{bmatrix}^T$ can be described by the Jones vector as follows[28]

$$\begin{bmatrix} \tilde{E}_x \\ \tilde{E}_y \end{bmatrix} = \frac{\tilde{t}_L + \tilde{t}_S}{2} \begin{bmatrix} 1 \\ \pm i \end{bmatrix} + \frac{\tilde{t}_L - \tilde{t}_S}{2} e^{\pm i2\theta} \begin{bmatrix} 1 \\ \mp i \end{bmatrix} \quad (2)$$

where $t_L$ and $t_s$ are frequency-dependent complex transmission coefficients (determine dispersion) when incident light is linearly polarized along its short and long axes, respectively. The rotation angle $\theta$ is defined as the angle between its symmetric axis and $x$-axis. The first and second terms are called polarization conserved and converted terms, respectively. Note that the $e^{\pm i2\theta}$ provides a knob to adjust phase by the rotation angle $\theta$ while not affecting the phase and dispersion of the polarization-conserved term. Now the goal is to ensure that, for both terms, the chosen nanostructures have to possess the same dispersion but different phases. Figure 2a, b show each nanostructure's (labeled from #1 to #8 as in Fig. 1b) phase delay of polarization-conserved and converted term, respectively. The phase delays are shown with respect to the phase delay of nanostructure #1, which is set as zero. The relative phase for other wavelengths is given in Fig. S4. These nanostructures are chosen from a large nanostructure library consisting of about 20,000 elements by considering their phases and dispersion properties, see Methods in Supplementary Information for details. In Fig. 2a, b, although the phase profiles show some deviation from the ideal linear shape, they increase monotonically and cover nearly $2\pi$. Contrarily, in the nanopillar case

(Fig. S4c), their phase profiles at blue and red wavelengths deviate from the ideal linear shape. It is worth mentioning that, based on calculation using scalar diffraction, a phase error of $\pm 0.15\pi$ introduced to the ideal linear phase profile (sampled by eight levels) given by Eq. 1 only lowers relative diffraction efficiency to 77% in the worst-case scenario (see Fig. S5). This shows that there exists a robustness against fabrication imperfections and coupling-induced phase errors. The similar dispersion among the eight nanostructures leads to broadband, high relative diffraction efficiency in measurement in Fig. 2c (see the blue curve). One can see that the dispersion-engineered metagrating has higher and more uniform curve across the design bandwidth from $\lambda = 450$ nm to 700 nm, while the conventional nanopillar grating (green curve) drops rapidly away from the peak. The dispersion-engineered metagrating has a peak value of 91.2%, and improvements of roughly 43% and 34% in relative diffraction efficiency at $\lambda = 450$ nm and 700 nm compared with the nanopillar metagrating. Their wavelength-averaged (from $\lambda = 450$ nm to 700 nm) relative diffraction efficiencies are 85.8% and 70.5%, respectively. It is worth mentioning that the dispersion-engineered metagrating can diffract light to +1 order for any incident polarization, although there is some efficiency variation because of different coupling between nanostructures under various incident polarization states. As shown in Fig. S6 for y-polarized incidence, the simulated averaged relative diffraction efficiency of the dispersion-engineered metagrating is up to 92.8%.

Figure 2d shows experimental results of diffraction efficiency (see Methods in Supplementary Information for measurement details) of which the dispersion-engineered metagrating and the nanopillar grating have wavelength-averaged (450 nm to 700 nm) diffraction efficiencies about 61% and 66%, respectively, yet the nanopillar grating has better peak diffraction efficiency.

## Dispersion-engineered chromatic metalenses

Indeed, the eight nanostructures of the dispersion-engineered metagrating can be used to implement other chromatic metasurface components, which exhibit significant broadband efficiency improvement compared with conventional metasurfaces comprising nanopillars. As an example, in Fig. 3a, we show a scanning electron microscope (SEM) image near the edge of a fabricated dispersion-engineered metalens (more SEM images are provided in Fig. S7). This metalens has a diameter of 250 μm and numerical aperture (NA) of 0.15 at $\lambda = 530$ nm. Such NA requires a minimal zone size (the radial distance over which the phase changes by $2\pi$) of about 3200 nm at the edge of the lens, which is equal to the dispersion-engineered metagrating's periodicity in Fig. 1b. One can see in Fig. 3b that the dispersion-engineered metalens has higher and flatter relative focusing efficiency compared with its counterpart of a nanopillar metalens with the same NA, whose peak relative focusing efficiency is 80%. Also, near the band edges at $\lambda = 450$ nm and 700 nm, the relative

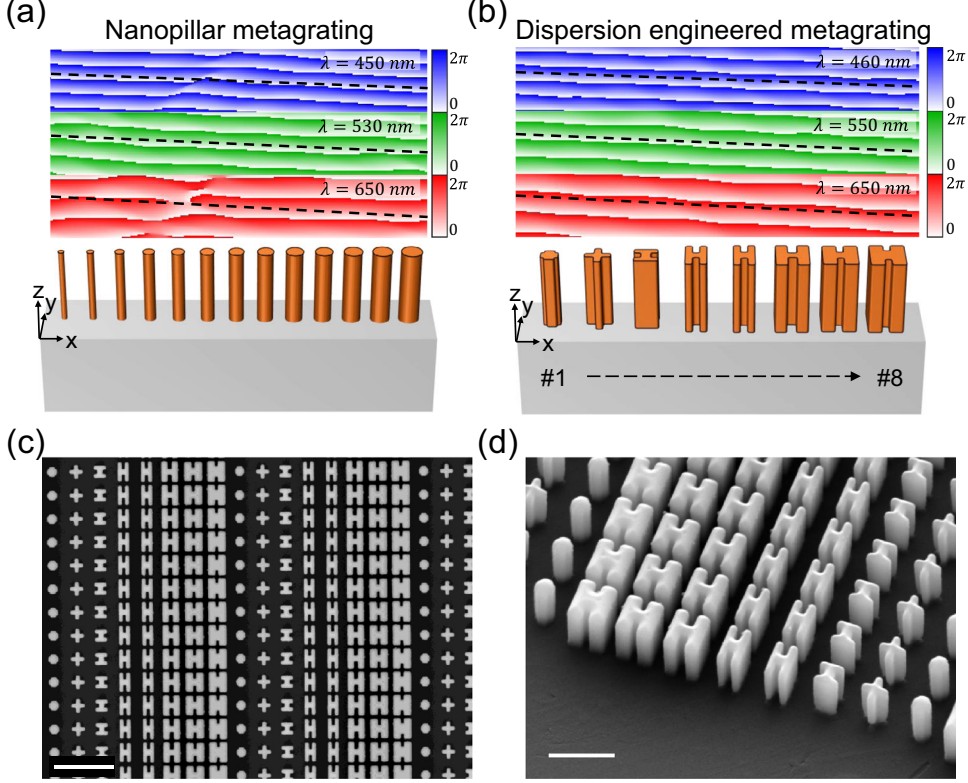

**Fig. 1 | Wavefronts of nanopillar and dispersion-engineered metagratings and scanning electron micrographs. a** A conventional nanopillar metagrating (designed for $\lambda = 530$ nm) and its simulated wavefront under illumination of three different wavelengths (450 nm, 530 nm, and 650 nm, respectively) in the visible. The simulated wavefront plot spans from 6 μm to 8 μm from the substrate surface in z-direction for each wavelength, respectively, and the color bars represent phase values in radians from 0 to $2\pi$. Different nanopillar diameters correspond to different dispersion properties, therefore the ideal linear wavefront (the black dashed line) can only be maintained within a small bandwidth and becomes distorted near the edges of the broadband chromatic response shown by the blue (450 nm) and the red (650 nm) wavefront plots. Some wavefront deviation at the design

wavelength could result from discrete phase sampling due to nanopillar's finite size and coupling between nanopillars. **b** A dispersion-engineered metagrating showing linear phase delay similar to that of the ideal ones at incident wavelengths of blue (460 nm), green (550 nm) and red (650 nm) in the visible. The shape and geometric parameters of the nanostructures are tuned in such a way that they can have very similar dispersion (i.e., the change of effective refractive index of the truncated waveguides) but different phases. For ease of discussion, nanostructures are labeled from #1 to #8. **c** and **d** Scanning microscope images of different views from a fabricated sample. The height of TiO$_2$ nanostructures is 600 nm. Scale bars are 1 μm and 500 nm, respectively.

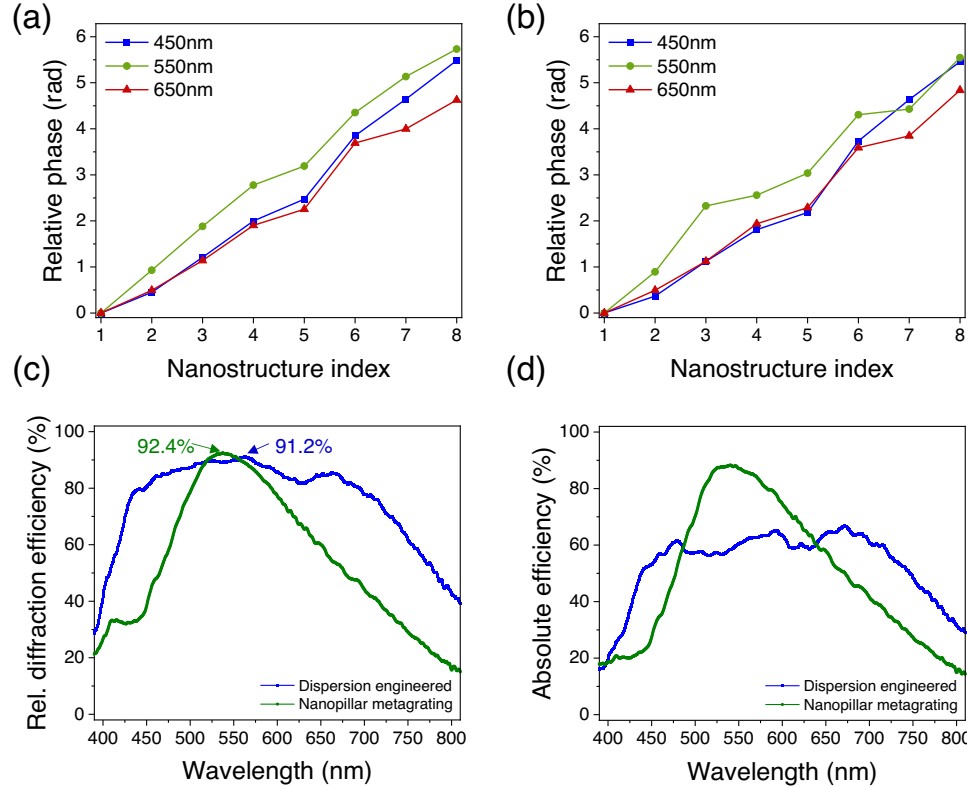

**Fig. 2 | Nanostructure phases and metagrating efficiency plots. a** and **b** Relative phase (at three wavelengths) of each nanostructure of the dispersion-engineered metagrating for the polarization conserved and converted terms, respectively. The *x*-axis shows the indices of the nanostructures depicted in Fig. 1b. The solid lines in **a** and **b** are to guide the eye. **c** Measured relative (Rel.) diffraction efficiency as a function of wavelength for the nanopillar (green) and dispersion-engineered (blue) metagratings shown in Fig. 1. This was measured in normal incidence under unpolarized illumination. The design bandwidth of the dispersion-engineered metagrating is from 450 nm to 700 nm. **d** Measured diffraction efficiency for the nanopillar and dispersion-engineered metagratings.

focusing efficiency increase by about 38% and 50%, respectively. Figure 3c shows the experimental absolute focusing efficiency of dispersion-engineered and nanopillar metalenses. In addition to having a near-flat curve, at $\lambda = 680$ nm, the dispersion-engineered metalens has a peak value about 72%, which is 42% higher compared with the nanopillar metalens. Next, we show another dispersion-engineered metalens of the same aperture size with a higher NA of 0.45. In such a case, the most outer zone of the dispersion-engineered metalens consists of three nanostructures only, as shown in Fig. 3d (more SEM images are provided in Fig. S8). Figure 3e shows the relative focusing efficiencies of the dispersion-engineered metalens and a nanopillar metalens of the same NA, whose peak relative focusing efficiencies are about 81% and 80%, respectively. In addition to the significant relative focusing efficiency improvement, one can also see that the relative focusing efficiency of the high-NA dispersion-engineered metalens is lower than that of the low NA one. This is because of Nyquist sampling[50]. Such Nyquist-sampling-related efficiency decrease is less significant for the nanopillar metalens because it is still well sampled (250-nm distance between two neighboring nanopillars) by about five nanopillars near the nanopillar metalens' edge. For absolute focusing efficiencies, one can refer to Fig. 3f. The dispersion-engineered and nanopillar metalenses have close averaged absolute focusing efficiency in the visible, although the nanopillar metalenses have higher peak values because of better transmission. Figure 3g, h show the focal spots and point spread function of the NA = 0.45 dispersion-engineered metalens, respectively. The focal spots are nearly diffraction-limited with high Strehl ratios in the range over the design bandwidth as shown in Fig. 3i (intensity profile of the focal spots can be found in Fig. S9). The focal spot images, point spread functions, Strehl ratios, and the focal

spots' intensity profiles of the NA = 0.15 dispersion-engineered metalens can be found in Fig. S10.

As an additional application example of our nanostructure set, in Fig. 4, we demonstrate a multifunctional metalens (NA = 0.15 at 530 nm) that can focus an incident plane wave to a donut spot with orbital angular momentum of 1. Figure 4a is an SEM image showing the center region of the fabricated metalens (More SEM images provided in Fig. S11). It has a spiral profile for generating an orbital angular momentum in the transmitted wavefront and a zone profile for focusing, which maintains nearly constant relative focusing efficiency of 70% from $\lambda = 450$ nm to 700 nm (Fig. 4b). The focused donut spots for selected wavelengths are shown in Fig. 4c, and their point spread functions are provided in Fig. S12.

## Discussion

It is worth mentioning that, although we utilized dispersion engineering, previous constraint about the product of lens diameter and NA in achromatic metalenses is not applicable[32,51]. This was because the dispersion-engineered metalenses and metagratings shown in this study are chromatic. We showcase the metalenses with a diameter of 250 um for ease of demonstration not because of the constraint. One can use recently developed large scale manufacturing[21–25,52] and use the same set of nanostructures to realize a large, broadband and efficient metalens.

To further improve the diffraction efficiency for the entire visible spectrum, it is important to increase relative diffraction efficiency near $\lambda = 400$ nm and also increase transmission over the entire visible bandwidth. These goals can be tackled by reducing the center-to-center distance between nanostructures. The rapid drop of relative diffraction efficiency of our dispersion-engineered metasurface

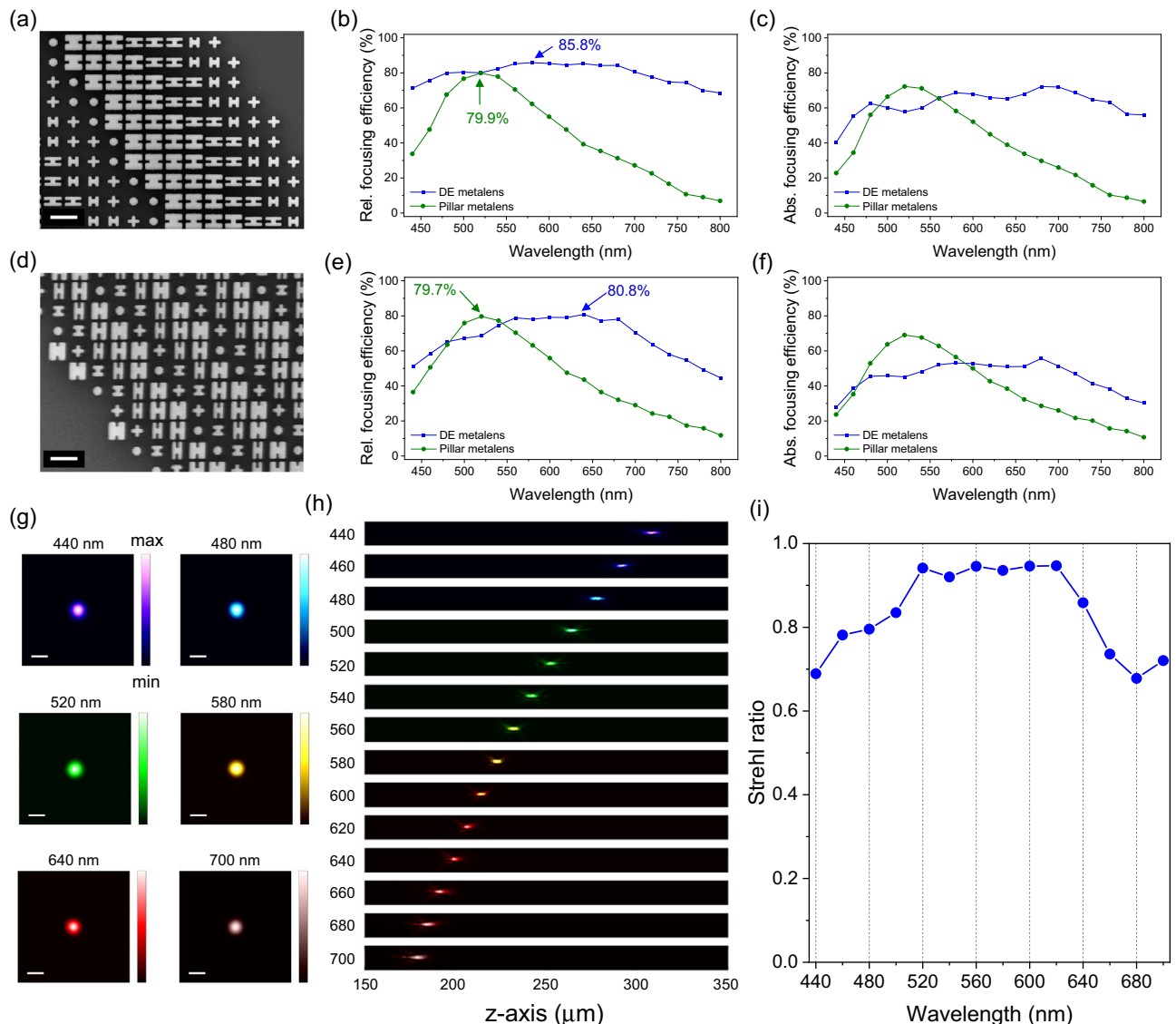

**Fig. 3 | Measurement results for dispersion-engineered metalenses. a** and **d** SEM images around the edge of fabricated dispersion-engineered metalenses (250 μm in diameter) of NA = 0.15 and NA = 0.45, respectively. Scale bars: 500 nm. These metalenses consist of the dispersion-engineered nanostructures shown in Fig. 1b. **b** and **e** Measured relative focusing efficiencies for the low and high NA metalenses (blue curves) in comparison with nanopillar metalenses (green curves) of the same corresponding NAs. For **b** and **e**, the labels mark their peak relative focusing efficiencies. **c** and **f** Measured absolute (Abs.) focusing efficiencies for the low and high NA dispersion-engineered (DE) metalenses. **g** and **h** Measured focal spots and point spread function for the dispersion-engineered metalens of NA = 0.45 for different wavelengths. Scale bars: 2 μm. Incident wavelength are labeled. **i** Measured Strehl ratio of the NA = 0.45 dispersion-engineered metalens. Data corresponding to NA = 0.15 dispersion-engineered metalens are provided in Supplementary Fig. S10.

components at blue wavelengths is due to the center-to-center distance of nanostructures used in this study is 400 nm. Therefore, when incident wavelength approaches 400 nm, higher order Bloch modes[53] and diffraction orders result in a decrease in transmission as well as the relative diffraction efficiency. Thus, to improve the performance at shorter wavelengths, a nanostructure library consisting of nanostructures with center-to-center distance smaller than 400 nm can achieve better transmission efficiency. We note that there are other combinations of dispersion-engineered nanostructures (see Fig. S13) showing similar diffraction efficiencies to that of Fig. 1c. This implies that there is room to shrink our nanostructure library. A nanostructure library consisting of nanostructures with smaller than 400 nm center-to-center distance can also have better overall transmission. In additional, we noticed that the dispersion-engineered metalenses have lower efficiencies than that of the metagrating. This can be improved by utilizing the recently developed grating averaging method[54–57],

where dispersion-engineered metagratings of different diffraction angles can be optimized for different metalens zones and stitched together afterwards to ensure high focusing efficiency.

The eight nanostructures in the metagrating design (Fig. 1) are selected from our nanostructure library (Fig. S2) built based on unit-cell approximation. Hence, mutual coupling between adjacent nanostructures exists and results in deviation from their unit-cell-approximated optical response when they are stitched to form a metagrating. We further show here that it is possible to tackle this challenge by fine-tuning their individual geometric parameters. The layout shown in Fig. 1b is used as an initial input for a particle swarm code paired with Lumerical (a commercial finite-difference time-domain solver) for optimization, where the lengths, widths and height of the eight nanostructures are set as variables. After optimization (Fig. 5a), the set of nanostructures shows 4% increase in the averaged (from $\lambda = 450$ nm to 700 nm) relative diffraction efficiency to 91.9%

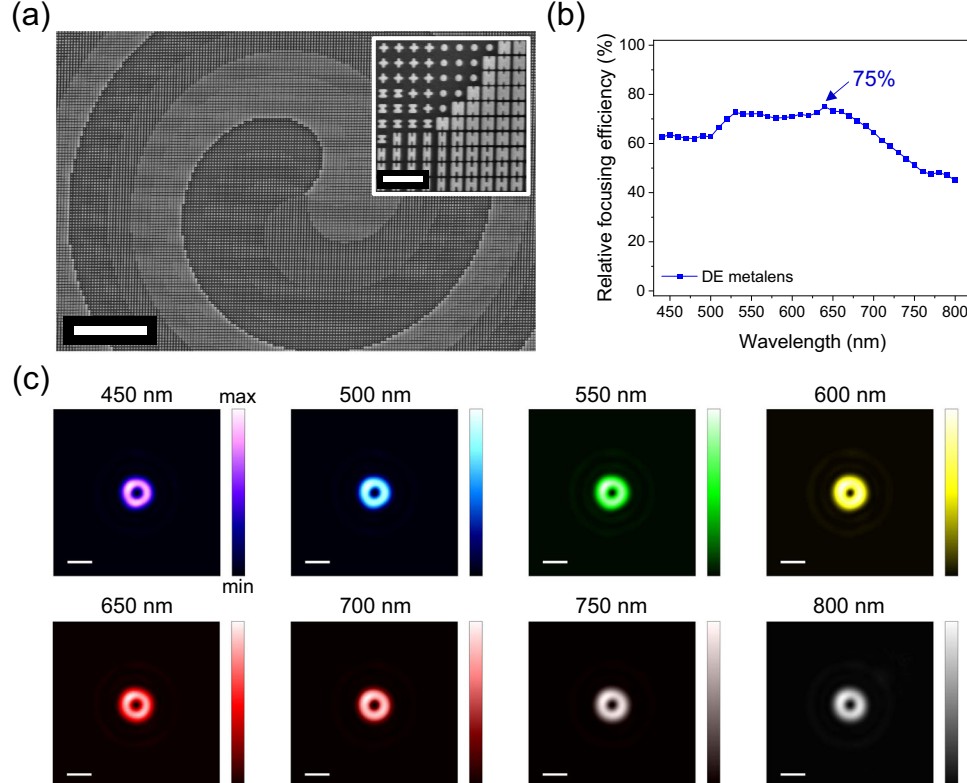

**Fig. 4 | Experimental results for a multifunctional dispersion-engineered metalens. a** SEM images taken around the center of the multifunctional dispersion-engineered metalens. The inset show a magnified image. Scale bars are 10 μm and 1 μm, respectively. The metalens was designed to focus an incident plane wave into a donut spot. **b** Measured relative focusing efficiency of the metalens. The label marks its peak relative focusing efficiency. **c** Focal spot profiles (false colored) for each incident wavelength labeled on the top of each image. Scale bars: 3 μm.

(red curve) with a peak value of 96%. Meanwhile, the averaged diffraction efficiency also increases by 4% (Fig. 5b) to 74.8%. The layout of the particle swarm optimized metagrating is given in Fig. S14a. In Fig. S14b, we also show its $0^{th}$ order efficiency in comparison with nanopillar metagratings. One can see that the power of the $0^{th}$ order is significantly suppressed over the design bandwidth for the dispersion-engineered metagratings. Such suppression is important for reducing imaging noises in many metasurface-based components such as metasurface spectrometers[38,58,59] and hybrid refractive-metasurface metalenses[37,60]. Furthermore, together with computation methods[61,62], our design can also be applied to realize achromatic images with increased image sharpness.

We have shown a set of dispersion-engineered nanostructures that have a similar dispersion property, i.e., their phase change over a wavelength region is nearly identical, while being able to impart different phases at a design wavelength in the visible. Considering these two factors in design results in a dispersion-engineered metagrating with near 90% relative diffraction efficiency averaged from $\lambda = 450$ nm to 700 nm. We further showcase that the same nanostructure set can be used to realize other metasurface components with high efficiency, including metalenses of NA = 0.15 and 0.45, as well as a multifunctional metalens capable of focusing plane wave into a donut-shaped focus. Our work indicates an approach to increase efficiency of metasurface components for numerous applications across industry and scientific research where having low zeroth order is important, for instance, in lithography, microscopy, endoscopy and virtual and augmented reality.

## Methods
### Design
We first created a 'library' of nanostructures through parameter sweep of mirror symmetric nanostructures. Finite-difference time-domain

(FDTD) simulations (Lumerical, Ansys) were performed to obtain far-field phase spectra for wavelengths 400–800 nm (in 10 nm per step) under circularly polarized incidence. Period boundary condition was used. The rotation angle of each element was kept constant during simulation as the rotation-induced phase delay can be predicted by Pancharatnam-Berry phase. After simulating each nanostructure, we obtained the polarization converted and conserved phases over the entire bandwidth from 400–800 nm ($\phi_{converted}$ and $\phi_{conserved}$) and fitted them by a quadratic polynomial ($\phi(\omega) = a_1\omega^2 + a_2\omega + a_3$, where $\omega$ is angular frequency and $a_1$ to $a_3$ are fitting coefficients) from wavelength 450 nm to 700 nm to get group delay and group delay dispersion. After the fitting, each nanostructure gets a number of goodness of fit. Subsequently, we removed those nanostructures whose goodness of fit is lower than 0.98 in order to exclude the nanostructures with strong resonances within the design bandwidth. Figure S2 shows the nanostructures, as blue circles located by their polarization converted and conserved phases at design wavelengths as x- and y-axis before and after applying the filter condition. The total amount of nanostructures reduced from about 20,000 to 7900. The black line in Fig. S2b indicates the target linear phases of grating (equally sampled 8 times from 0 to $2\pi$). The next goal is to make sure each target black circle can find a corresponding blue one. Note that, because phase is relative, the blue circles are allowed to shift together horizontally and vertically until a figure of merit is maximized or minimized. To implement such process, we used Matlab to write a two-variable (phase shifts along x- and y-directions) particle swarm code by using the 1st-order grating diffraction efficiency or wavefront aberration function as figure of merit. For the former figure of merit, the 1st-order diffraction efficiency was calculated by Fourier transform and scalar diffraction theory, then the data was sent back to the Matlab particle swarm algorithm as feedback for the code to find the best shifts. In the latter,

(a)

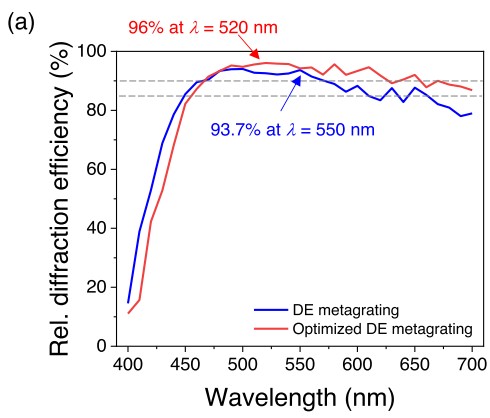

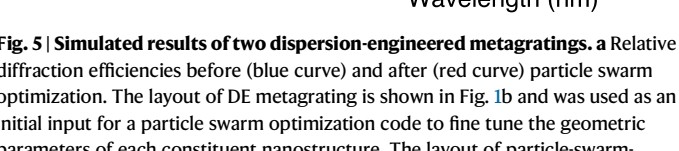

(b)

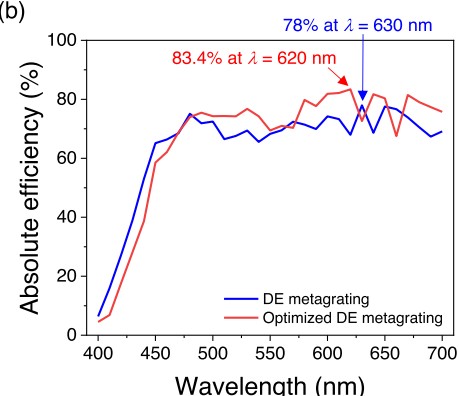

**Fig. 5 | Simulated results of two dispersion-engineered metagratings. a** Relative diffraction efficiencies before (blue curve) and after (red curve) particle swarm optimization. The layout of DE metagrating is shown in Fig. 1b and was used as an initial input for a particle swarm optimization code to fine tune the geometric parameters of each constituent nanostructure. The layout of particle-swarm-optimized one can be found in Fig. S14. The relative (Rel.) diffraction efficiency is averaged under *x*- and *y*-polarized incidence. The dashed lines label 85% and 90% and the color arrows point peak values. Both metagratings were designed from $\lambda = 450$ nm to $\lambda = 700$ nm. **b** Corresponding diffraction efficiencies to for the metagratings.

we used nanostructure phases from the library and calculated the wavefront aberration function with respect to the target linear one. It is worth mentioning that the above process does not guarantee the global best solution because of approximations. We therefore repeated the above-mentioned process and generated ~30 different metagrating designs. The best one was reported in the main text while a few of the rest are shown in Fig. S13.

### Fabrication
A 600 nm thick electron beam resist film (ZEP520A, Zeon Specialty Materials Inc.) on a 0.5 mm thick, 1-inch diameter, double side polished fused silica (JGS2) substrate is prepared by spin-coating and baking. We note that the thickness of the resist film defines the height of the metasurface. After coating the resist film surface with charge dissipation material (ESPACER, Showa Denko), the metasurface pattern is written into the resist film using a 125 kV electron beam (ELS-F125, Elionix Inc.) with 1 nA current and 5 nm beam step size. The e-beam resist is then developed in chilled o-Xylene (puriss.p.a., ≥99.0% (GC), Sigma Aldrich) followed by IPA rinse and $N_2$ blow-dry, creating holes of the metasurface pattern in the resist film with vertical sidewalls. The metasurface patterned resist film is then conformally coated with amorphous $TiO_2$ using low-temperature atomic layer deposition (Savannah 200, Cambridge Nanotech), until the holes are completely filled. The excessively grown $TiO_2$ layer is then dry etched with inductively coupled plasma reactive ion etching (PlasmaPro 100 Cobra 300, Oxford Instruments) with $CHF_3/Ar/O_2$ plasma, until the resist layer is exposed. Then, the exposed resist layer is removed with downstream plasma ashing (Matrix Plasma Asher, Matrix Systems Inc.) at 220 °C, leaving only the 600 nm tall $TiO_2$ metasurface structure on the fused silica substrate.

### Measurement
We built an optical setup to measure metagrating's spectrum in a single shot. A broadband incoherent light source (EQ-99X LDLS, Energetiq) and a reflective-mirror collimator (RC08APC-P01, Thorlabs) were used together with a lens (#59-875, Edmund Optics) to focus light on a metagrating. The transmitted light was collected by a high NA = 0.95 Olympus objective (MPLAPON100x, Olympus). An achromatic tube lens was used to relay the back focal plane of the objective (where rainbow grating orders can be seen) to the entrance aperture of an integrating sphere (IS200, Thorlabs) equipped with a spectrometer (SILVER-Nova, StellarNet Inc.). An iris was placed in between the objective and the tube lens as a spatial filter to block unwanted light.

The positions of the integrating sphere were adjusted such that the power spectra of the 1st-order diffraction and transmitted light can be measured, respectively. When measuring transmitted light, the integrating sphere was brought close to the tube lens such that all transmitted grating orders can pass through the entrance pupil of integrating sphere. The relative diffraction efficiency was then calculated by dividing the measured values of power. For diffraction efficiency, we moved the sample away from the light path and measured power spectrum as reference. For metalens measurement, we built a microscope using the NA = 0.95 Olympus objective paired with a tube lens (focal length $f = 180$ mm). A schematic diagram can be found in ref. 43.

### Data availability
The data that support the findings of this study are available from the authors on reasonable request.

### Code availability
The code used in this study are available from the authors on request. The layouts and simulation files of the dispersion-engineered and nanopillar metagratings can be downloaded from the authors websites at https://capasso.seas.harvard.edu/ and https://www.weitingchen-meta.com/.

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

## Acknowledgements
This work was supported by the Defense Advanced Research Projects Agency (Grant No. HR00111810001) and by the Air Force Office of Scientific Research under award Number FA9550-21-1-0312. This work was performed in part at the Harvard University Center for Nanoscale Systems (CNS); a member of the National Nanotechnology Coordinated Infrastructure Network (NNCI), which is supported by the National Science Foundation under NSF award no. ECCS-2025158.

## Author contributions
W.T.C. and F.C. conceived the study. J.P. fabricated the samples. W.T.C., J.M., S.M., and K.M.Y. performed simulations and developed codes. W.T.C. and J.P. measured the samples and analyzed the data. All authors wrote the manuscript, discussed the results, and commented on the manuscript.

## Competing interests
The authors declare no competing interests.
