## [Peer Review File · Nature Communications]

Dispersion-engineered Metasurfaces Reaching Broadband 90% Relative Diffraction EfficiencyREVIEWER COMMENTS

Reviewer #1 (Remarks to the Author):

The manuscript reports the design and experimental demonstration of metagratings and metalenses with excellent zeroth order suppression across the visible spectrum. Overall the manuscript is well-written and the simulation approach and experimental data are clearly presented. I recommend publication of the manuscript after the following comments are properly addressed.

- 1) The definition of diffraction efficiency: the commonly adopted convention refers to diffraction efficiency as the fraction of the incident optical power that appears in a single diffraction order. In this paper the authors define diffraction efficiency as the the power of the +1 grating order divided by the power of transmitted light. While I agree that a high +1/0 order power ratio is more important for reducing veiling glare in many application scenarios, the unconventional definition can be misleading to the readers, especially those who are not familiar with the field. I recommend the authors to adopt the conventional definition and revise the title to 'Dispersion-engineered Metasurfaces for broadband zeroth order suppression'.
- 2) The devices described in this paper are chromatic, which the authors also allude to in the Discussion section. This is an important point that should be made clear in the abstract. The manuscript can also be further strengthened if the authors can provide a few concrete examples of potential applications involving chromatic diffractive elements such as those discussed in the manuscript.
- 3) Plots showing a) the axial intensity distribution of metalens focal spots vs. wavelength; and b) the Strehl ratio vs. wavelength (Fig. S6) should be added to Fig. 3.
- 4) Can the authors comment on variations of the device performances across the spectral range due to mutual coupling between neighboring meta-atoms?
- 5) The comparison of nanopillar metagrating and the dispersion-engineered metagrating assume different meta-atom pitches for the two. Can the authors compare two gratings with the same pitch? This would allow full 2π phase coverage at the longer wavelength end in the nanopillar metagrating.

Reviewer #2 (Remarks to the Author):

Chen et al. present their work on dispersion-engineered metasurfaces to obtain a broadband operation with high diffraction efficiency. To obtain their goal, they designed meta-atoms with nearly identical dispersion properties but different phases for the transmitted light. Based on a library of structures they selected 8 different geometries to realize different kinds of metasurfaces and demonstrated experimentally their properties.

Overall, it is nice to see what kind of potential metasurfaces and the correct design of their unit-cells can have. However, I see this work as one step of many to improve the properties of metasurfaces further without introducing new aspects in the research. The amount of 20,000 elements that were calculated is impressive but also shows the problem of the work: the incremental improvement of efficiency values and bandwidth requires huge computational resources for a specific material system and wavelength area. It does not mean I do not appreciate the work the authors have done here (which is great) but I cannot see the innovation of the work compared to other improvement steps in the past. Like in their previous work (see Fig 4C in PNAS 113 (38) 10473-10478) the authors showed already a broadband behavior for birefringent meta-atoms with efficiency values similar to the

presented values in this work, which is also covering 450-750 nm bandwidth. Therefore, I do not think that the work has the impact and the broad interest to be published in Nature Communications. I recommend a more optics specific journal where this work might find the right readership.

The manuscript is written in a clear manner but I think it requires some improvement for the discussion of the chromatic behavior of the metalenses and the consequences for focusing light. What also is not discussed by the authors is the angular dependence of the diffraction efficiency for all the different meta-atoms. I could imagine that for larger scattering angles the completely different designs result in a strong variation of scattering efficiency. This might be also the reason why the NA0.45 metalens falls back behind the NA0.15 metalens in its focusing efficiency.

Reviewer #3 (Remarks to the Author):

Chen and Park's work presents a broadband meta-grating with high efficiency through dispersion engineering. By selecting meta-elements with similar dispersion profiles, the authors were able to achieve a grating structure that maintains high efficiency across the visible wavelength range of 450 to 700 nm. Furthermore, three additional metalenses (two focusing lenses and a multifunctional lens) were demonstrated using the principles proposed in this work. While the innovative nature and quality of this research make it suitable for publication in Nature Communications, I recommend publication upon major revisions to address several areas that need improvement.

#1 The authors should be more careful in their use of the term "diffraction efficiency." While a paragraph at the end of the Introduction section clarifies that their definition of diffraction efficiency refers to relative efficiency, rather than absolute diffraction efficiency, the title and abstract are still confusing and potentially misleading in their use of the term. Upon reading the title and abstract, I initially assumed that "diffraction efficiency" referred to absolute diffraction efficiency. Therefore, it is important to make corresponding changes to the title and abstract to clearly communicate the intended meaning of "diffraction efficiency."

#2 In this work, the authors compare their dispersion-engineered meta units with a conventional design based on circular pillars. They have carried out a thorough search of a library and optimized their dispersion-engineered design to achieve good broadband performance with similar dispersion for all meta units. On the other hand, the dispersion of the circular pillars in the conventional design can vary depending on design details such as height, period, and radius. The authors did not discuss whether the performance shown in Fig. 1(a) is representative of typical conventional designs, raising the question of whether the comparisons in this work are fair. It would be helpful if the authors could address this issue and provide further discussion on this point.

#3 In Equation 2, please define the angle "theta".

#4 The representation of Figs. 2(a) and 2(b) could be improved by including a plot of the combination of the polarization conserved and converted phase, which determines the dispersion of the meta unit according to Eq. 2. Additionally, a plot of the relative phase for the different circular pillars in Fig. 1(a) at three different wavelengths would provide a more comprehensive understanding of the dispersion-engineered meta unit.

#5 In this work, the broadband performance is demonstrated within the wavelength range of 450 to 700 nm. It would be useful to understand the limiting factors that prevent the extension of the wavelength range beyond these boundaries. Is it possible to further extend the range and, if so, what measures could be taken to achieve this?

#6 The authors compare their dispersion-engineered metasurface with a conventional metasurface using circular pillars in all figures except for Fig. 4. It is curious that no such comparison is provided in this case. It would be useful to know what the curve in Fig. 4(b) would look like for a conventional metasurface design and why a comparison was not included in this figure.

#7 In Fig. S3(b), several curves differ significantly from the others. It is difficult to determine which wavelengths they correspond to due to the color scheme used in this figure. Despite these divergent curves, the dispersion-engineered meta grating appears to perform well across the 450 to 700 nm bandwidth. This suggests that the grating design has some robustness with regard to the dispersion of each meta unit, allowing for good broadband performance even when the dispersion profiles of the individual elements are not entirely similar. However, the authors have not provided a quantitative analysis or discussion on this point. It would be helpful if they could include such a discussion to give readers a better understanding of how similar the dispersion profiles of the individual elements need to be in order to achieve broadband performance when adapting this method for designing their own meta devices.

Reply to reviewers' comments

Reviewer #1 (Remarks to the Author):

... I recommend publication of the manuscript after the following comments are properly addressed.

1) The definition of diffraction efficiency: the commonly adopted convention refers to diffraction efficiency as the fraction of the incident optical power that appears in a single diffraction order. In this paper the authors define diffraction efficiency as the power of the +1 grating order divided by the power of transmitted light. While I agree that a high +1/0 order power ratio is more important for reducing veiling glare in many application scenarios, the unconventional definition can be misleading to the readers, especially those who are not familiar with the field. I recommend the authors to adopt the conventional definition and revise the title to 'Dispersion-engineered Metasurfaces for broadband zeroth order suppression'.

Reply: Actually, we defined diffraction efficiency in an early paragraph of our manuscript because we noticed that the definition of diffraction efficiency is ambiguous. For instance, in *Science* **345**(6194), 298-302 (2014) and Goodman's book "Introduction to Fourier Optics", the term was defined with respect to the power of transmitted and incident light, respectively. Also, Yakov G. Sосkind's "Field Guide to Diffractive Optics" from SPIE defines "diffraction efficiency" and "relative diffraction efficiency" as a fraction of the incident power contained within the order and with respect to a perfectly reflecting or transmitting substrate, respectively. Therefore, to make the terms clear, we revised the title of our manuscript to "Dispersion-engineered Metasurfaces Reaching Broadband 90% **Relative** Diffraction Efficiency". Also, we have added the definition of both diffraction efficiency (with respect to incident light) and relative diffraction efficiency (with respect to transmitted light) in the abstract.

2) The devices described in this paper are chromatic, which the authors also allude to in the Discussion section. This is an important point that should be made clear in the abstract. The manuscript can also be further strengthened if the authors can provide a few concrete examples of potential applications involving chromatic diffractive elements such as those discussed in the manuscript.

Reply: We have modified the abstract accordingly. References of potential applications for aberration correction, spectroscopy and computational imaging based on our dispersion-engineered metasurfaces have been added in the discussion section.

3) Plots showing a) the axial intensity distribution of metalens focal spots vs. wavelength; and b) the Strehl ratio vs. wavelength (Fig. S6) should be added to Fig. 3.

Reply: We have added the plots accordingly.

4) Can the authors comment on variations of the device performances across the spectral range due to mutual coupling between neighboring meta-atoms?

Reply: Due to the use of unit cell approximation, there is a mutual coupling between neighboring meta-atoms, which lowers efficiency. In our original submission, we tackled this challenge by using an application programming interface to link our nanostructure library and FDTD simulation

package Lumerical in such a way that a full-wave simulation can be performed and provides feedback to select the eight nanostructures. This was reported in detail in the Supplementary Information. In the revision, we further use a particle swarm code to tackle the challenge and to improve relative diffraction efficiency. The result has been added as a new figure to the main text as Fig. 5 and discussed in the last paragraph in the discussion session.

5) The comparison of nanopillar metagrating and the dispersion-engineered metagrating assume different meta-atom pitches for the two. Can the authors compare two gratings with the same pitch? This would allow full 2π phase coverage at the longer wavelength end in the nanopillar metagrating.

Reply: We have added the comparison into Fig. S14. The 250-nm pitch metagrating has better diffraction efficiency compared with the 400-nm one. They both have similar relative diffraction efficiencies over the design bandwidth, which are much lower than the dispersion-engineered metagratings.

Reviewer #2 (Remarks to the Author):

... It does not mean I do not appreciate the work the authors have done here (which is great) but I cannot see the innovation of the work compared to other improvement steps in the past. Like in their previous work (see Fig 4C in PNAS 113 (38) 10473-10478) the authors showed already a broadband behavior for birefringent meta-atoms with efficiency values similar to the presented values in this work, which is also covering 450-750 nm bandwidth.

Reply: Actually, in our original manuscript, we had discussed why our dispersion engineered metasurface is superior (the first paragraph on page 4) compared with those metasurfaces implemented by Pancharatnam-Berry phase. The meta-atoms in the PNAS are nanofins which can only diffract, in unpolarized incidence, less than 50% of incident light to +1 order due to the conjugated Pancharatnam-Berry phase. Our dispersion-engineered metagrating is polarization insensitive (Fig. S6), which is not capped by the 50% limit. Additionally, in the PNAS, there are three designs for red, green and blue wavelength regions and none of them show broadband high efficiency as our dispersion-engineered one. In the revision, we have added Fig. S1 as a comparison between the dispersion engineered metagrating and the ones in the PNAS paper.

... The manuscript is written in a clear manner but I think it requires some improvement for the discussion of the chromatic behavior of the metalenses and the consequences for focusing light.

Reply: Incidentally, Reviewer 1 raised a similar question. We have made changes in the abstract to emphasize that the dispersion-engineered metasurfaces are chromatic. In the revision, we also discussed promising applications based on our chromatic dispersion-engineered metasurfaces in the paragraph before conclusion.

What also is not discussed by the authors is the angular dependence of the diffraction efficiency for all the different meta-atoms. I could imagine that for larger scattering angles the completely different designs result in a strong variation of scattering efficiency. This might be also the reason why the NA0.45 metalens falls back behind the NA0.15 metalens in its focusing efficiency.

Reply: As pointed out, the reason about the efficiency decrease in NA=0.45 metalens is due to sampling. In such NA, there are only 3 nanostructures (in case of 400 nm distance between two

neighboring nanostructures) in a zone, while the $NA = 0.15$ one corresponds to 8 nanostructures. According to scalar diffraction, the less the nanostructures, the lower the efficiency. In our original submission, this together with solutions were discussed in the second paragraph in the discussion section.

Reviewer #3 (Remarks to the Author):

... While the innovative nature and quality of this research make it suitable for publication in Nature Communications, I recommend publication upon major revisions to address several areas that need improvement.

#1 The authors should be more careful in their use of the term "diffraction efficiency." While a paragraph at the end of the Introduction section clarifies that their definition of diffraction efficiency refers to relative efficiency, rather than absolute diffraction efficiency, the title and abstract are still confusing and potentially misleading in their use of the term. Upon reading the title and abstract, I initially assumed that "diffraction efficiency" referred to absolute diffraction efficiency. Therefore, it is important to make corresponding changes to the title and abstract to clearly communicate the intended meaning of "diffraction efficiency."

Reply: In the revision, we have changed the term in title to "relative diffraction efficiency" and have made it clear by adding its definition in the abstract. In revision, we follow the convention that diffraction efficiency is defined with respect to the power of incident light.

#2 In this work, the authors compare their dispersion-engineered meta units with a conventional design based on circular pillars. They have carried out a thorough search of a library and optimized their dispersion-engineered design to achieve good broadband performance with similar dispersion for all meta units. On the other hand, the dispersion of the circular pillars in the conventional design can vary depending on design details such as height, period, and radius. The authors did not discuss whether the performance shown in Fig. 1(a) is representative of typical conventional designs, raising the question of whether the comparisons in this work are fair. It would be helpful if the authors could address this issue and provide further discussion on this point.

Reply: The 250-nm-pitch and 600-nm-height nanopillars are chosen according to our previous publication (Nano Lett. 16, 7229-7234 (2016)). It represents a faithful benchmark regarding the performance of nanopillar metasurfaces as one can see that its measured peak absolute efficiency is as high as $\sim 90\%$ (Fig. 2d). In the revision, we have added Fig. S14 (simulated efficiencies of dispersion-engineered and nanopillar metagratings) with another 400-nm pitch (the same pitch as the dispersion-engineered one) nanopillar metagrating for reference.

#3 In Equation 2, please define the angle "theta".

Reply: We have added the definition into the third line below equation 1.

#4 The representation of Figs. 2(a) and 2(b) could be improved by including a plot of the combination of the polarization conserved and converted phase, which determines the dispersion of the meta unit according to Eq. 2. Additionally, a plot of the relative phase for the different circular pillars in Fig. 1(a) at three different wavelengths would provide a more comprehensive understanding of the dispersion-engineered meta unit.

Reply: Because the polarization conserved and converted terms correspond to orthogonal polarized states and don't interfere, we show their phases separately in Fig. S3. In the same figure, we have added the nanopillars' phases. In addition, to help readers easily understand and replicate the results and parameters of interest, we will provide all Lumerical simulation files and layouts on the authors' websites.

#5 In this work, the broadband performance is demonstrated within the wavelength range of 450 to 700 nm. It would be useful to understand the limiting factors that prevent the extension of the wavelength range beyond these boundaries. Is it possible to further extend the range and, if so, what measures could be taken to achieve this?

Reply: Yes, it is possible. In the second paragraph in the discussion section, we have identified how to extend the range. To extend towards the blue wavelength, the unit cell size needs to be reduced from 400 nm to a smaller one, while towards the red, the nanostructure height needs to increase a little from 600 nm. According to your comment, another dispersion engineered design is shown in Fig. 5 as well as Fig. S14 and discussed in the paragraph before the conclusion section showing how particle swarm optimization can be used to increase efficiency further.

#6 The authors compare their dispersion-engineered metasurface with a conventional metasurface using circular pillars in all figures except for Fig. 4. It is curious that no such comparison is provided in this case. It would be useful to know what the curve in Fig. 4(b) would look like for a conventional metasurface design and why a comparison was not included in this figure.

Reply: The nanopillar vortex lens would have similar efficiency performance to that in Fig. 3. Also, in Fig. 4, we would like to emphasize that the dispersion-engineered elements can also be used to achieve multifunctionality (focusing and changing orbital angular momentum). Therefore, in the initial design of experiment, we didn't include the nanopillar counterpart.

#7 In Fig. S3(b), several curves differ significantly from the others. It is difficult to determine which wavelengths they correspond to due to the color scheme used in this figure. Despite these divergent curves, the dispersion-engineered meta grating appears to perform well across the 450 to 700 nm bandwidth. This suggests that the grating design has some robustness with regard to the dispersion of each meta unit, allowing for good broadband performance even when the dispersion profiles of the individual elements are not entirely similar. However, the authors have not provided a quantitative analysis or discussion on this point. It would be helpful if they could include such a discussion to give readers a better understanding of how similar the dispersion profiles of the individual elements need to be in order to achieve broadband performance when adapting this method for designing their own meta devices.

Reply: We have made the y-axis longer so that the figure becomes clearer. The computation cost is too high to perform a Monte Carlo analysis for robustness study. Instead, in Fig. S5, we used scalar diffraction and Fourier transform to estimate how much phase error is tolerable. It shows that, for the phase of each nanostructure, a phase error of about $\pm 0.15\pi$ is acceptable.

REVIEWERS' COMMENTS

Reviewer #1 (Remarks to the Author):

All of my comments have been satisfactorily addressed. I now recommend publication of the manuscript.

Reviewer #2 (Remarks to the Author):

In their revised version the authors addressed the technical comments that I made. However, I am still not convinced that the impact of the work is sufficiently high to warrant a publication in Nature Communications. The authors argue about the PB phase metasurfaces that are not polarization insensitive compared to their design. But there also exist other polarization insensitive dielectric metasurfaces in the VIS that work broadband and with efficiencies of up to 80% (see for example ACS Appl. Mater. Interfaces 2022, 14, 31, 36019–36026). Of course, the authors demonstrate an improvement compared to the state of the art but I see this improvement as rather incremental than as a breakthrough. Here, I will leave it to the editor to make a final decision based on all review reports.

Reviewer #3 (Remarks to the Author):

In this revised manuscript, the authors addressed all my comments and I have no further questions or comments regarding this manuscript. I support its publication in Nature Communications.

Reply to reviewers' comments

Reviewer #1 (Remarks to the Author):

All of my comments have been satisfactorily addressed. I now recommend publication of the manuscript.

Reply: Thank you for your recommendation.

Reviewer #2 (Remarks to the Author):

In their revised version the authors addressed the technical comments that I made. However, I am still not convinced that the impact of the work is sufficiently high to warrant a publication in Nature Communications. The authors argue about the PB phase metasurfaces that are not polarization insensitive compared to their design. But there also exist other polarization insensitive dielectric metasurfaces in the VIS that work broadband and with efficiencies of up to 80% (see for example ACS Appl. Mater. Interfaces 2022, 14, 31, 36019–36026). Of course, the authors demonstrate an improvement compared to the state of the art but I see this improvement as rather incremental than as a breakthrough. Here, I will leave it to the editor to make a final decision based on all review reports.

Reply: We have cited this work in Ref. 48. Actually, this is another example similar to the PB metasurfaces in the PNAS paper in previous comment. According to Fig. 2g in the paper, its efficiency at 450 nm is only 10%, while our metasurface possesses about 80% and 60% in relative diffraction efficiency and diffraction efficiency, respectively. The efficiency bandwidth of the metasurface shown in the paper is obviously not comparable to our dispersion engineered metasurfaces.

Reviewer #3 (Remarks to the Author):

In this revised manuscript, the authors addressed all my comments and I have no further questions or comments regarding this manuscript. I support its publication in Nature Communications.

Reply: Thank you for supporting the publication of our manuscript.